# Community cervical cancer screening: Barriers to successful home-based HPV self-sampling in Dabat district, North Gondar, Ethiopia. A qualitative study

Bikila Soboka Megersa[1]*, Hermann Bussmann[2], Till Bärnighausen[1], Achenef Asmamaw Muche[3], Kassahun Alemu[3], Andreas Deckert[1]

1 Heidelberg Institute of Global Health, Heidelberg University, Heidelberg, Germany, 2 Department of Applied Tumor Biology, Institute of Pathology, Heidelberg University, Heidelberg, Germany, 3 Department of Epidemiology and Biostatistics, Institute of Public Health, University of Gondar, Gondar, Ethiopia

* bikila070@gmail.com

**Data Availability Statement:** All relevant data are within the manuscript and Supporting Information files.

## Abstract

### Objective

To explore the barriers to successful home-based human papillomavirus (HPV) self-sampling in North Gondar, Ethiopia.

### Methods

The study participants were women who had previously participated in a community-wide home-based HPV self-sampling pilot study, community health workers, women's development army leaders, and the sample collectors of the home-based HPV self-sampling pilot study. A community based qualitative descriptive study was conducted. We applied purposive and convenience sampling. In total, 47 women participated in the study (in-depth interviews n = 22, four focus group discussions n = 25, 6–7 participants each). The study employed thematic analysis for clustering the emerged themes.

### Results

Husband disapproval was identified as the main barrier to the acceptance of home-based HPV self-sampling. Social influence, lack of knowledge about cervical cancer and screening, lack of health education on cervical cancer and HPV-based screening, feeling healthy, and religious influence were identified as additional barriers. Fear of using Evalyn brush® for self-sampling was found to be the main barrier to the provision of a quality sample. The inability of the sample collectors to check the proper utilization of Evalyn brush® and the difficulty in understanding the instructions did also contribute to the low-quality. Providing health education concerning cervical cancer and HPV self-sapling to women, male involvement in the screening program, and linking the screening service to existing local health facilities were suggested to guarantee the success of home-based HPV self-sampling.

**Funding:** The project was funded by the Open Society Foundation as part of the Civil Society Leadership Award for a master thesis to BSM. In addition, we acknowledge financial support by the Baden-Württemberg Ministry of Science, Research and the Arts and by Ruprecht-Karls-Universität Heidelberg. The funders had no role in study design, data collection and analysis, decision to publish, or preparation of the manuscript.

**Competing interests:** The authors have declared that no competing interests exist.

## Conclusions

Educating women regarding cervical cancer and HPV testing, providing clear instructions on how to collect self-sample, and male involvement in the screening program are prerequisites for a successful implementation of home-based HPV testing. Women empowerment should also be focused to overcome the identified sociocultural barriers. Furthermore, the screening program should guarantee the timely provision of the test results and offering women follow-up examinations and treatment for abnormal findings.

## Introduction

Cervical cancer ranks as the fourth most commonly diagnosed and cause of cancer death among women globally [1]. This disease ranks as the second leading female cancer for both incidence and mortality in Ethiopia [2, 3]. It is estimated that 31.5 million women aged $\geq$ 15 years are at risk for cervical cancer in Ethiopia [4]. In Ethiopia, the number of cervical cancer deaths are on the rise and a growing proportion of women are diagnosed at advanced stages [5].

In contrast to high-income countries where population-based vaccination, organized cytology-based screening, and treatment programs have successfully reduced cervical cancer incidence and mortality, the burden has remained unchanged in low- and middle-income countries (LMICs) due to poor health care services [6–8].

The World Health Organization (WHO) recommends the use of either visual inspection with acetic acid (VIA) or HPV test strategies for cervical cancer screening (CCS) in LMICs, since cytology-based screening requires too many resources [9]. HPV-based screening is considered to be the most suitable strategy to increase the coverage of CCS in low-resource settings by enabling women to take a self-collected cervicovaginal sample for HPV testing at home [6, 10–16]. Also, studies conducted in low-income settings have shown that population based self-collected HPV testing followed by treatment for HPV-positive women has the potential to be a cost-effective screening strategy [17, 18].

The identified drivers of HPV self-sampling cost-effectiveness include the level of increase in cervical cancer screening attendance, lower HPV self-sampling materials and testing costs, and attracting under-screened women [19–21]. Additionally, HPV-based screening provides greater protection from invasive cervical cancer and it has a higher sensitivity compared to cytology-based testing [12, 22–24]. Moreover, HPV-based CCS helps to significantly improve the acceptance of CCS in low-resource settings [25–27].

As part of the implementation of the national cancer control strategy in 2015, the Ethiopian Federal Ministry of Health (FMoH) introduced HPV vaccination for 14-years-old girls in 2018 and has planned to provide vaccination for girls of 9–14 years through the routine immunization program [28, 29]. Besides this, the FMoH has planned to achieve 80% cervical cancer screening coverage for women 30–49 years using VIA by the year 2020. Additionally, the FMoH encourages the adoption of the HPV DNA testing in order to detect HPV precancerous cervical lesions through community-wide screen and treat approaches [28]. However, the current CCS coverage of 2.9% in Ethiopia is still unacceptably low [30].

Several barriers limit the acceptance and success of CCS in LMICs, including limited knowledge of cervical cancer, lack of awareness about screening methods, misconceptions about causes, ignorance of prevention methods, spousal disapproval, misbelief of not being at risk, fear of self-harm and contravening religious and/or cultural obligations of modesty during the screening procedure [31–34].

In order to increase CCS coverage in Ethiopia, a community-wide pilot study was conducted in Dabat district, North Gondar in 2017, utilizing a home-based HPV self-sampling approach. Women aged 25–65 years were offered Evalyn brush® for HPV self-sampling in their homes. The findings of this study showed that the acceptance rate for the HPV self-sampling was 85%. Although the study management followed strictly international guidelines and procedures to assure the quality of samples, about one-fifth of the samples were insufficient for HPV testing [35]. However, the acceptance and provision of an adequate sample were unexpectedly low compared to other studies conducted in Africa. HPV self-sampling was accepted 93% in Nigeria [26], 98.5% in rural Senegal [36], 98% in rural Madagascar [37], and almost all the collected samples were adequate for HPV testing [26, 36, 37].

Prior to the pilot study, a first qualitative study was conducted in a similar setting that identified the perceptions, acceptability, barriers, and preferences for HPV self-sampling in focus group discussions (FGDs) [38]. Complementary, our study aimed to explore the actual barriers to home-based HPV self-sampling after the home-based HPV self-sampling campaign, and to identify the implementation challenges of the pilot study. We extended the qualitative data collection tools and included interviews with screened women.

## Materials and methods

### Study setting

University of Gondar referral hospital is the only tertiary hospital in the North Gondar Zone, providing services for about 5 million population; it has been providing CCS via VIA and cryotherapy services since 2013 [39]. Chila Kebele, where the participants were selected, is one of 30 kebeles (smallest administrative unit) in the Dabat district of the North Gondar Zone. The only health facility in Chila kebele is a primary satellite health post, run by community health workers (CHWs) [40]. Additionally, the women's development army leaders (WDALs; a group of volunteer 'model woman' work under the supervision of CHWs) mainly participate in community mobilization and awareness creation towards improving maternal and child health [41, 42].

### Study design

A qualitative study design was used to explore the barriers to successful home-based HPV self-sampling, using in-depth interviews (IDIs) and FGDs. We conducted IDIs to explore participants' knowledge, perceptions, beliefs, and experiences on barriers to acceptance of home-based HPV self-sampling and to the provision of a sufficient quality HPV testing sample as explained in the literature [43–45]. Additionally, we conducted FGDs in order to explore the knowledge, perceptions, and beliefs towards cervical cancer, screening and the barriers to the acceptance of self-sampling in the society [44, 46].

### Study participants and sampling process

Study participants were women who had participated in a community-wide home-based HPV self-sampling, CHWs, WDALs, and the sample collectors of the pilot study. All IDIs and two FGDs were conducted with women who participated in home-based HPV self-sampling pilot study. Additionally, one FGD targeted the HPV sample collectors to grasp their experiences with instructing the women on the self-sampling instrument (Evalyn brush®), and another FGD was conducted with CHWs and WDALs, considering their in-depth understanding of the cultural context and the community's health-seeking behaviour [47, 48].

A purposive sampling technique was used to select the women, based on HPV testing sample quality criteria. CHWs, WDALs, and the sample collectors were selected by convenience

sampling based on their working schedule, distance from FGD site, and availability to participate [49]. Each of the four FGDs comprised 6–7 participants. The sample size for the in-depth interview was determined based on data saturation [50].

## Data collection instruments and procedures

Semi-structured interview guides were developed for both IDIs and FGDs in English and then translated to Amharic. The questions were prepared by the study management of the pilot study, taking into consideration the literature on CCS in Ethiopia [34, 51, 52]. The health belief model theoretical framework also guided the development of the questions, considering many factors affect acceptance of CCS, including beliefs which could differ across cultures [53–55].

The IDI guide was pretested on five women, whereas the FGD guide was pretested on a group of four. Afterwards, the interview guides were adjusted accordingly. The interview guide for the FDG with the HPV sample collectors was checked by the Dabat research site coordinator. None of the women who participated in the pretest were included in the actual data collection.

Data collection was done within two weeks in June 2019. In light of the sensitive issue, the IDIs and FGDs were carried out by three female qualitative data collectors, experienced as data collectors in the community (Dabat research center) and in the pilot study. Yet, the FGD with the HPV testing sample collectors was facilitated by the principal investigator at the Dabat health center. The other FGDs were conducted at a health post in Chila kebele and the IDIs took place at the respondents' households in a confidential setting. A tape-recorder was used to capture all data from the IDIs and the FGDs. Additionally, field notes were taken for all FGDs to capture the main themes of the discussion.

The main points of IDI discussions were knowledge, perceptions, and beliefs on cervical cancer and screening, the barriers to acceptance of home-based HPV self-sampling and to provide a sufficient quality HPV testing sample. In contrast, the FGDs discussions focused on the knowledge, perceptions, and beliefs on cervical cancer and screening. The sample collector's discussion guide focused on the barriers to acceptance of home-based HPV self-sampling in the society and the provision of a sufficient quality HPV testing sample (see S1 File).

## Data analysis

The audio-recorded data were transcribed into Amharic. Then, the consistency of the transcription was checked by randomly choosing parts of the recorded data and crosschecking with the transcription. Thereafter, the transcriptions were translated into English and coded into main themes using NVivo (version 12). We employed thematic analysis for identification of the four main themes, based on the participants' experience and opinions [56, 57]: i) knowledge, perception, and beliefs toward cervical cancer and screening; ii) barriers to acceptance of HPV self-sampling; iii) barriers to the provision of a sufficient quality HPV testing sample; and iv) suggestions of the participants to guarantee the success of home-based HPV self-sampling.

## Ethics statement

Ethical approval for the study was obtained from the Ethics Committee of Heidelberg University School of Medicine, Germany, ethical approval number: S-132/2019 and the Institutional Review Board of the University of Gondar, Ethiopia, ethical approval number: V/P/RCS/05/2032/2019. The information sheet was explained verbally to the participants until they have completely understood it. Information concerning the objectives of the study, voluntary participation, and the right to withdraw from participation at any point of the study, handling of

personalized data, possible risk and benefits of the participation in the study was given to the participants. After the participant's questions were addressed and they confirmed the information they received was clear, written consent was obtained from all the study participants.

## Results

### Sociodemographic characteristics of the respondents

Data were collected from a total of 47 participants (25 (53%) participated in FGDs, 22 (47%) in IDIs). The average age was 36 years. The majority of the participants were married (83%) and had no formal education (45%). The average family had 4 children (Table 1).

### A) Lack of knowledge

Most of the respondents had heard about cervical cancer and it is known as *Nekersa* (cancer) in the community, but some of the participants understood cervical cancer as haemorrhoids (*Kintarot)*, uterine prolapse, and fistula. Some of the participants indicated that they had heard about cervical cancer for the first time when the pilot study was implemented. However, some of pilot study's participants acknowledged that they did not know about cervical cancer despite their participation in HPV self-sampling (see Table 2 quote 1).

Foul discharge from the uterus, which is believed to cause 'shrinking' of the woman's cloth on her buttock, and irregular bleeding were the most commonly mentioned symptoms of cervical cancer. However, one IDI participant did not know any symptoms. Most of the respondents referred the effect and symptoms of cervical cancer to the uterus (see Table 2 quote 2).

None of the participants mentioned HPV as the causative organism of cervical cancer (see Table 3). However, majority of them believed that exposing the genitalia to evaporation coming from urinating on the hot ground *(Gerefta)*, sudden exposure of the body to sunlight *(Mitat)*, smoke from open fire cooking (Tis) and home delivery as the primary causes of cervical cancer (see Table 2 quote 3).

**Table 1. Sociodemographic characteristics of IDI and FGD participants in Dabat district, North Gondar, Ethiopia.**

| Variables* | Total participants | Sufficient sample | Insufficient sample | CHWs and WDALs* | Pilot study sample collectors |
|---|---|---|---|---|---|
| Total n (%) | **47 (100.0)** | **16 (34.0)** | **19 (40.4)** | **6 (12.8)** | **6 (12.8)** |
| **IDIs** | 22 (46.8) | 10 (62.5) | 12 (63.2) | | |
| **FDGs** | 25 (53.2) | 6 (37.5) | 7 (36.8) | 6 (100.0) | 6 (100.0) |
| Age, mean ± SD | 36 ± 9 | 33 ± 2 | 39 ± 2 | 40 ± 3 | 28 ± 4 |
| Marital status, n (%) | | | | | |
| **Married** | 39 (83.0) | 16 (100.0) | 16 (84.2) | 5 (83.0) | 2 (33.3) |
| **Single** | 4 (8.5) | 0 (0.0) | 0 (0.0) | 0 (0.0) | 4 (66.7) |
| **Divorced** | 1 (2.0) | 0 (0.0) | 1 (5.3) | 0 (0.0) | 0(0.0) |
| **Widowed** | 3 (6.4) | 0 (0.0) | 2 (10.5) | 1 (17.0) | 0 (0.0) |
| Educational status, n (%) | | | | | |
| **No formal education** | 21 (44.7) | 11 (68.7) | 10 (52.6) | 0 (0.0) | 0 (0.0) |
| **Primary education** | 17 (36.2) | 3 (18.7) | 9 (47.4) | 5 (83.0) | 0 (0.0) |
| **Secondary education** | 3 (6.4) | 2 (12.5) | 0 (0.0) | 1 (17.0) | 0 (0.0) |
| **Tertiary education** | 6 (12.8) | 0 (0.0) | 0 (0.0) | 0 (0.0) | 6 (100.0) |
| Number of children, mean ± SD | 4 ± 1 | 4 ± 1 | 5 ± 1 | 3 ± 1 | 1 ± 1 |

*IDIs: In-depth interviews, FGDs: Focus group discussions, SD: Standard deviation, CHWs: Community health workers, WDALs: Women's development army leaders.

**Table 2. List of quotes from the IDIs and FGDs of all four groups of participants.**

| Nr. | Group * | Quote |
|---|---|---|
| 1 | A | "I don't even know about this disease (cervical cancer); I don't have any word to say about it, this disease has come recently." (ESIDI01) |
| 2 | A | "[. . .] When this disease affects the uterus, it bleeds and sometimes produce whitish discharge which can be visible on a woman's dress (Kemis) and causes to shrink on her buttocks. It (cervical cancer) also causes severe pain during the menstrual cycle." (ESIDI09) |
| 3 | D | "We get this disease by exposure to evaporation coming out when we urinate on the hot ground (Gerefta) and a sudden exposure to sunlight (Mich). [. . .] When we go to a market, we urinate on the hot ground and sit on hot stones, this is what brings cervical cancer to us. We also think it can also be caused during delivery as we give birth at home." (HEDA04) |
| 4 | B | "I don't think the modern treatment can identify and treat this disease, once a woman got the disease, it is all about her, her fate will be death. I have never heard someone treated and cured of this disease. It is really dangerous." (ESFGD06) |
| 5 | A | "In my opinion, all of us can get this disease, but we don't know why it only manifest on some women, I think it is due to God's protection that some of us don't show symptoms." (ISIDI09) |
| 6 | B | "It (cervical cancer) affects a woman's life as it leaves a uterus without function, she could no more carry a baby in her womb. This is almost equal to death; can you imagine what life will be without having a baby." (ISFGD06) |
| 7 | A | "I really felt bad about it. First, I thought it was easy, but the experience was not easy. [. . .] I was embarrassed to take the sample from my body (genitalia). To be honest, I did it for the sake of respecting the girls (sample collectors) who come to my home walking all that long distance. I was not totally happy about it." (ISIDI07) |
| 8 | D | "I don't see any benefits of this screening when I'm healthy and don't have any symptoms of the disease; so, why I need be screened?" (HEDA02) |
| 9 | A | "[. . .] later, some of the residents of our local community insulted me for participating in the screening and everybody was blaming me because I said yes to those girls (sample collectors); I was really embarrassed for giving that sample to those girls." (ISIDI07) |
| 10 | A | "I didn't tell my husband about my participation. He told me not to participate in the screening. Even he refused the girls who were collecting the sample to enter our compound. [. . .] But it was bad if he would have known as I gave the sample, he could even beat me." (ISIDI 11) |
| 11 | C | "There are misconceptions in the community; some women were not willing to participate particularly wives of religious leaders were not involved in the screening program." (DC02) |
| 12 | A | "I didn't face a challenge to collect the sample, but you know we have used this instrument for the first time, and I felt stress. I was also afraid to use the instrument considering many things people were talking about it." (ESIDI06) |
| 13 | C | "It wasn't easy to show the procedure of self-sampling for some of the women; we spent a lot of time to make them understand how to use it (the self-sampling device). Some of them forgot every step of the procedure immediately after they went to their bedrooms to collect the sample," (DC01) |
| 14 | C | "[. . . .] after we explained all the procedures to the women, they went to their bedrooms to sample themselves. We were told not to open and check the sample once the women closed and gave it to us; so, we collected a closed instrument from them, and we can't be sure whether a woman has used it or not." (DC03) |
| 15 | A | "Our husbands also need to learn and support us. They need to be educated on this issue (CCS), how to support their wife to participate in the screening program and give support in case she is sick from this disease." (ESIDI08) |
| 16 | C | "[. . .] many women did not receive their results and those who identified having the disease were not linked to follow-up and treatment center. Currently, we feel shame to meet the community members as we have promised them to bring back their results and to link them to follow-up and treatment center in case, they are diagnosed with the disease during sample collection." (DC02) |

\* A: In-depth interviews with participants of the pilot study, B: Focus group discussion with participants of the pilot study, C: Focus group discussion with HPV sample collectors of the pilot study, D: Focus group discussion with community health workers and women's development army leaders.

Most of the participants believe that cervical cancer can be prevented by avoiding the mentioned causes and that cervical cancer could be identified and treated at an early stage in a health facility. Nevertheless, some perceived screening as a diagnosis to seek treatment after

**Table 3. Knowledge, perception, and beliefs of IDI and FGD participants on cervical cancer and screening in Dabat district North Gondar, Ethiopia.**

| Themes | Study participants | | |
|---|---|---|---|
| | **Screening participants with sufficient quality sample (IDIs and FGD)** * | **Screening participants with insufficient sample (IDIs and FGD)** | **Community health workers and women's development army leaders (FGD)** |
| Knowledge about cervical cancer (CC)* | Majority associated CC with foul discharge and irregular bleeding and explained it as an incurable disease. | | |
| | To some, CC was known as Nekersa (cancer), fistula, uterine prolapse, and hemorrhoids (Kintarot). | | |
| | These two groups had less understanding about CC. | | This group had more understanding of CC. |
| | Some of the participants did not have detailed knowledge about the female reproductive tract, could not identify the cervix from uterus. | Most of the participants did not have detailed knowledge about the female reproductive tract, could not identify the cervix from uterus. | |
| Beliefs about the causes and prevention of cervical cancer | Commonly mentioned causes: smoke from open fire cooking (Tis), sudden exposure of the body to sunlight (Mitat), evaporation coming out from urinating on the hot ground (Gerefta), and home delivery. | | |
| | None of the participants mentioned HPV as the cause of CC. | | |
| | Additional causes: early marriage, sitting on hot chair or stone, multiple pregnancies, teenage pregnancy, and husband's lack of attention to the woman's wellbeing. | Additional causes: bad spirits, lack of personal hygiene, wearing cloths exposed to sunlight for a long time, having sexual intercourse outside of wedlock, early marriage, and lack of proper nutrition. | Additional causes: lack of personal hygiene during the menstrual cycle, holding urine for long a period (Shintmat), having multiple sexual partners and multiple pregnancies. |
| | The participants believe avoiding the above-mentioned causes prevents CC. | | |
| Knowledge about cervical cancer screening. | Majority suggested that women should be screened for CC at any time regardless of having symptoms. | | |
| | Some want to be screened only when they get sick or have some of the CC symptoms. | | |
| | | One woman was only willing to be screened when the disease interrupts her normal daily functions. | |
| Perception towards susceptibility, severity of CC and benefits of participating in CC screening. | All assumed any woman can get sick from CC. | | |
| | All perceived CC as very severe disease and some said once women get sick from the disease, it is difficult to get cured. | | |
| | They explained the severity of the disease in terms of foul discharge, irregular bleeding, social exclusion, mental impact, inability to give birth, and care for a family. | | One participant witnessed a woman being excluded from society due to foul discharge. |
| | Many recognized the benefits of their participation in screening. | | |
| Belief in the early identification and treatment of cervical cancer. | Majority believe that CC can be identified and treated at a health facility. Some believe that CC is incurable with modern medicine. | | |
| | Some think that CC can be treated with holy water (Tsebel). | Some mentioned using holy water (Tsebel) alone or with modern treatment can cure the disease. | One participant thought the disease can be treated and cured with holy water (Tsebel) for those who belief in God. |

* IDIs: In-depth interviews, FGD: Focus group discussion, CC: Cervical cancer, HPV: Human papillomavirus.

having symptoms or before they are severely sick. On the other hand, some believe that the disease cannot be treated and cured at a health facility. Whereas other women took the view that cervical cancer could be treated at both health facilities and by religious interventions, particularly by applying holy water (*Tsebel*) (see Table 2 quote 4 and the summary in Table 3).

## B) Perception towards severity of cervical cancer, and benefits of HPV self-sampling

The participants shared the common perception that every woman is at risk of cervical cancer and they related their susceptibility to the reasons they mentioned as the causes of the disease (see Table 2 quote 5). Among all participants, cervical cancer was perceived as a very severe disease. They mentioned the inability to become pregnant and give birth, the difficulty to care for the own family, divorce as a consequence of the disease, and psychological impact from foul discharge due to stigmatization, and either self-exclusion from social interaction or exclusion by the society (see Table 2 quote 6).

The majority had recognized the benefits of their participation in HPV self-sampling (see Table 3). The most perceived benefit was identifying themselves as free from cervical cancer. Nevertheless, some of the participants had only done the test to do a favor to the sample collectors while one participant mentioned not having benefited from her participation because she felt providing a self-collected sample breaks cultural obligations to be modest (see Table 2 quote 7).

## C) Barriers to acceptance of HPV self-sampling and to the provision of a sufficient quality HPV test sample

**Lack of proper health education/training.**   Some participants of IDIs and the sample collectors highlighted that health education was not properly provided prior to the implementation of the pilot study, most importantly on how to properly use Evalyn brush® (see summary in Table 4). They also considered the lack of health education as one of the reasons that drove to the misconceptions among some of the community members that the Evalyn brush® could impregnate women, which resulted in some of the husbands' disapproval.

**Lack of symptoms/feeling healthy.**   Although some stated that they are willing to participate in HPV self-sampling if they get an opportunity, others said that they would only participate if they have symptoms or are sick (see Table 2 quote 8).

**Social influence.**   According to the sample collectors some of the women wanted to know whether their neighbours were participating in self-sampling before deciding to take part. Similarly, some of the women reported feeling embarrassed about participating in HPV self-sampling because their neighbours were not happy with their participation. Likewise, some were influenced by the misconception in the community regarding the use of Evalyn brush® (see Table 2 quote 9).

**Husband disapproval.**   Husband disapproval as a barrier was brought up in some of the IDIs and the FGD with sample collectors. Some of the participants did not disclose their

**Table 4. Barriers to acceptance of HPV self-sampling and to the provision of a sufficient quality HPV test sample identified by IDI and the sample collectors FGD participants.**

| Themes | Study Participants | | |
|---|---|---|---|
| | Screening participants with sufficient quality sample in-depth interviews | Screening participants with insufficient sample in-depth interviews | Sample collectors focus group discussion |
| Barriers to acceptance of HPV self-sampling | Main barriers mentioned: husband disapproval | | |
| | Other main barriers: lack of health education on CC and HPV self-sampling and feeling of embarrassment to discuss with a spouse. | Other main barriers: some felt it is culturally embarrassing to perform self-sampling and to discuss with their husbands. | Other main barriers: social disapproval, lack of awareness about CC and HPV self-sampling, misconceptions about consequences of using Evalyn brush®, religious influence, and time constraints. |
| | Fear of the consequences of using the self-sampling device and feeling healthy. | | |
| | Some mentioned community disapproval and fear of what others could say about their participation in HPV self-sampling. | | |
| Barriers to the provision of a sufficient quality HPV testing. | Main barriers mentioned: fear of hurting their genitalia with the self-sampling device. | | Barriers: fear of using the Evalyn brush® and the inability of the sample collectors to check the proper utilization of Evalyn brush®. |
| | Some mentioned difficulty in understanding the procedure instructions to perform self-sampling. | | Difficulty for some women to understand the procedure and use of Evalyn brush®. |
| | Some afraid of becoming pregnant from using the self-sampling device. | Lack of providing proper instructions on how to use Evalyn brush®. | |
| Suggestions of the participants to guarantee the success of home-based HPV self-sampling. | Commonly mentioned suggestions: male involvement in the screening program, community health education about CC and HPV self-sampling, and linking those with abnormal findings to follow-up and treatment. | | |
| | Other suggestions: the sustainability of the screening program and linking the service to the local health facility. | Another suggestion: the provision of test results on time. | |
| | | | Other suggestions: providing training on how to use Evalyn brush® and making self-sampling kit available at primary health care level. |

participation to their husbands since they perceived self-sampling breaks the social obligations of women's modesty. Moreover, the sample collectors reported that some of the women who participated in self-sampling without getting permission from their husbands were threatened. One sample collector discussed that she could not meet a woman for HPV self-sampling consultation since a woman's husband seriously impeded her not to meet his wife. However, the majority of women reported that they had discussed self-sampling with their husbands and that they received their support (see Table 2 quote 10).

**Religious influence on the acceptance of home-based HPV self-sampling.** Although most women indicated that their religion did not influence their participation, the sample collectors felt that religion influenced the acceptance of home-based HPV self-sampling. Most of the women who were married to religious leaders and those who took part in consubstantiation were not willing to participate in self-sampling. These women justified their withdrawal with the interdiction to use Evalyn brush® and they perceived it could defile them. Some of the respondents also stated the instrument was new in the area and they did not know whether it was acceptable or not according to their religion (see Table 2 quote 11).

## D) Barriers to provide a quality self-sample for HPV testing

**Fear of using the Evalyn brush®.** According to the sample collectors the fear of using Evalyn brush® was one of the reasons why some of the women could not provide a quality self-collected sample. Some of the participants mentioned that they were afraid to apply the self-sampling instrument due to lack of proper training, feeling of embarrassment, and fear of unknown consequences due to misconceptions in the community. According to the sample collectors, some of the women were emotionally instable while getting instructed. Though most of the women thought that the test would be painful, only one participant reported a painful genital self-collection (see Table 2 quote 12).

**Challenges to collect the self-sample.** The majority of the participants indicated that the instructions were comprehensible. However, the sample collectors mentioned that some of the women had faced challenges to understand and apply the Evalyn brush® after receiving the instructions; some of the collectors went to women's private bedrooms to show them how to properly stand and perform the sampling procedure (see Table 2 quote 13).

**The inability of the sample collectors to check the proper utilization of Evalyn brush®.** The sample collectors were not sure whether the women who provided a low-quality sample had used Evalyn brush® properly or not, because the sample collectors were not trained to check the brush after use (see summary Table 4, and Table 2 quote 14).

## E) Suggestions of the participants to guarantee the success of home-based HPV self-sampling

The participants suggested that health education should be given on cervical cancer and on the benefits of participating in HPV self-sampling. Also, they stressed the importance of providing proper training on how to use Evalyn brush®. In addition, the sample collectors suggested that misconceptions in the community towards the use of Evalyn brush® should be addressed. Since the lack of men involvement played a significant role, the participants suggested health education and explaining the procedure to spouses together (see Table 2 quote 15).

Some of the participants desired to link the HPV self-sampling service to existing local health facilities to guarantee the continuation of the program and the accessibility to women whom the sample collectors might miss to meet. Also, they indicated that women could receive their results when they go to a health facility for other purposes. Thus, they perceive this could help them to overcome sociocultural barriers to participate in the screening.

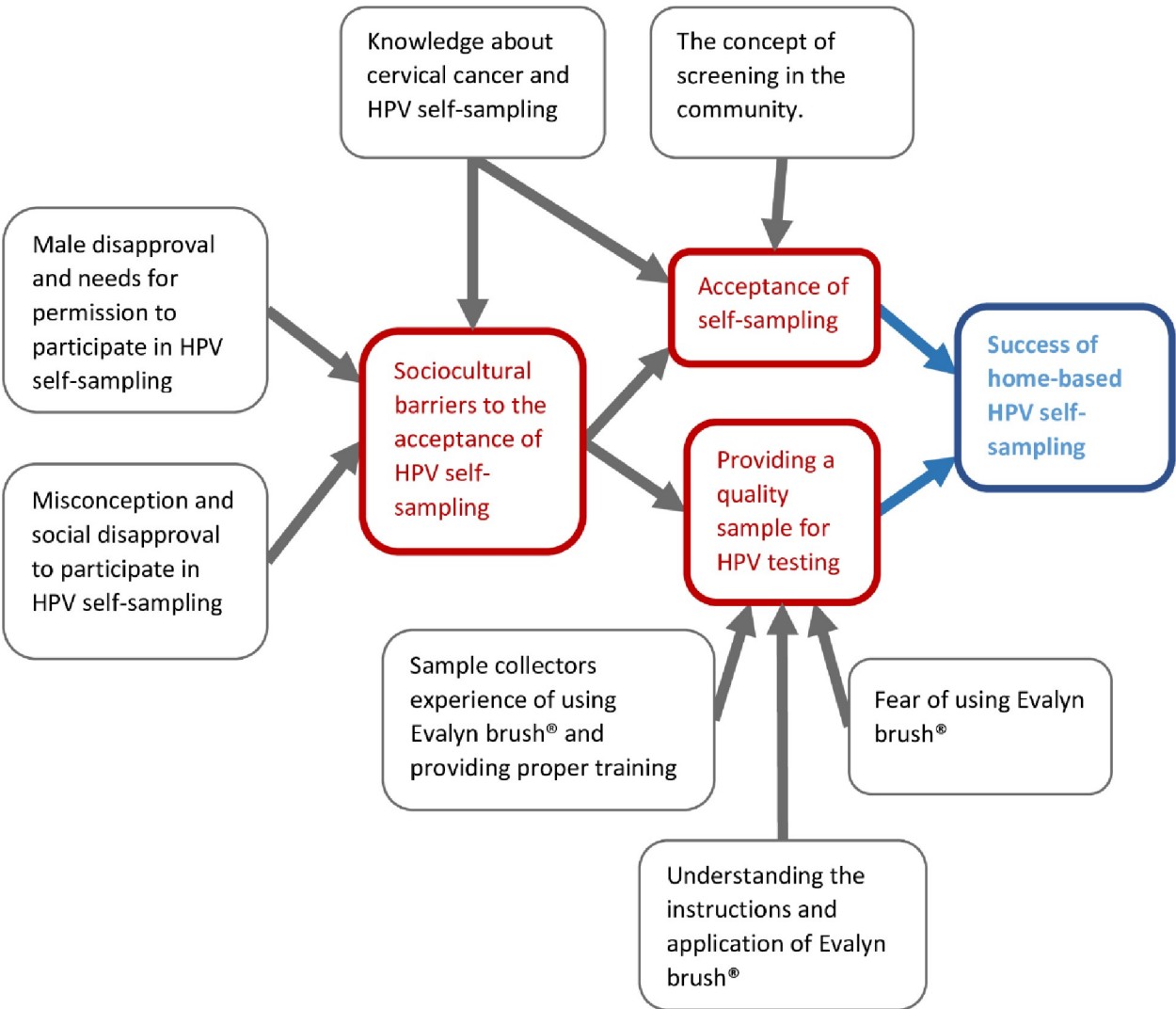

**Fig 1. The factors influencing the success of HPV self-sampling in Dabat district, North Gondar, Ethiopia.**

Most of the women who did not provide a sufficient sample for HPV testing reported that they had never received their screening results and some women had not being offered follow-up and treatment for abnormal findings. They stated that they are not willing to participate again in the screening and do not want to recommend the screening to other women. The participants mentioned that this could be one of the potential barriers to the success of future the screening program (see Table 2 quote 16).

Altogether, the success of home-based HPV self-sampling mainly depends on the degree to which the screening is accepted in the community and the quality of the collected samples. In turn, the acceptance and the quality of the collected samples are affected by several additional factors (suggested theoretical model see Fig 1).

## Discussion

Our study identified two key components for successful home-based HPV self-sampling in rural Ethiopia: the degree to which the screening is accepted in the community and the quality

of the collected samples for the HPV testing (see Fig 1). Husband disapproval was identified as the main barrier concerned with the acceptance of HPV self-sampling. Social influence, lack of knowledge about cervical cancer and screening, lack of health education on cervical cancer and HPV-based screening, feeling healthy, and religious influence were identified as additional barriers to the acceptance of home-based HPV self-sampling. Fear of using Evalyn brush® for self-sampling was found to be the main barrier to the provision of a quality sample. The inability of the sample collectors to check the proper utilization of Evalyn brush® and the difficulty in understanding the procedure instructions did also contribute to the low-quality samples.

Husbands' disapproval relates to the patriarchal society in Ethiopia where women are not allowed to participate in sexual and reproductive health services without consulting their husbands; women are expected to be modest [58]. It might also be due to men's interest in controlling their partner's use of reproductive health services or that the women themselves do not want to participate in self-sampling and use their partner's refusal as a justification for non-participation [59]. In general, men's negative attitudes towards CCS and disapproval of women's participation limit the acceptance of CCS [15, 59, 60]. Involving men in CCS programs, therefore, is essential to facilitate women's participation and decrease barriers to CCS.

We discovered that a strong relationship among the neighbours seems to be one of the reasons why some of the women were interested in checking their neighbours' participation and felt embarrassed when their neighbours do not endorse their participation. Limited knowledge about cervical cancer and screening probably contributes to this social disapproval. On the other hand, several studies found that close contacts such as family members, friends, and neighbours had a positive influence on the acceptance of cervical screening as they might discuss their fears and doubts before actually participating in the program [59–61]. These contrary findings could result from the low level of awareness about cervical cancer and screening in our study. In line with that, Austad et al, (2018) revealed that community level suspicion about the harmful consequences of participating in screening was one of the barriers to the acceptance of CCS.

Another sociocultural barrier to the acceptance of HPV self-sampling was religious influence. Similar to the findings of our study, religion has been identified as one of the barriers to the acceptance of HPV self-sampling in other African countries [15, 62], and in Australia [63].

The study participants had limited knowledge about cervical cancer and screening although they have been educated in the pilot study to some extent. The fact that more than half of the respondents of the pilot study did not have formal education could have contributed to the limited knowledge. In general, limited knowledge about causes, symptoms, prevention methods of cervical cancer, and screening have been identified as barriers by several other studies in Ethiopia [51, 52, 64–67] and elsewhere in Africa [31, 33, 68, 69]. For instance, some participants of a study conducted in Nigeria did not have information about cervical cancer [33], and some only had limited knowledge about the treatment of cervical cancer [31]. Likewise, misconceptions such as sudden exposure to sunlight (Mitat), urinating in a dirty area when it is sunny, bad spirits and poor personal hygiene as the causes of cervical cancer were discovered by others; and similarly, none of the interviewees of other studies mentioned HPV as the aetiology of cervical cancer [34, 38].

The belief that cervical cancer could not be treated and cured could be linked to the behaviour of rural women in Ethiopia to visit health facilities at the advanced stage of the disease only when the choices of effective treatment are already very limited [5]. However, having knowledge about cervical cancer and HPV testing seems to increase the acceptance of genital self-sampling [59, 70].

Lack of proper health education contributed to the refusal of HPV self-sampling in the pilot study. Interventional studies conducted in African countries have demonstrated that providing health education on cervical cancer and HPV significantly increases acceptance of self-sampling [71–73]. Also, providing education and information about HPV testing, and

educating the women on how to carry out HPV self-sampling before they participate in a self-sampling program increases the acceptance of self-screening [59, 74–77]. Therefore, the provision of health education, with support of clear educational materials on cervical cancer, causes, prevention methods, treatment options, and Evalyn brush® would help to address this limited knowledge. Additionally, women need to be given information on HPV self-testing and be trained on how to perform self-sample.

Similar to our findings, studies in Ethiopia and Kenya have revealed the common perception that any woman could get cervical cancer and as a very severe disease [34, 68, 78–80]. Most of the women recognized the benefits of taking part in the screening program and held the opinion that cervical cancer can be detected at an early stage and be treated [64, 68, 78–80]. Likewise, stigmatization and social exclusion of a woman having the disease from society has been reported in Ethiopia [34, 38].

Feeling healthy due to lack of symptoms has also been identified as the primary [78] and the second most common reason for not participating in CCS methods such as VIA in Ethiopia [51]. This could be due to a lack of understanding of the screening and the nature of cervical cancer [67, 81]. Additionally, some women in another Ethiopian study tended to give priority to diseases that present symptoms [51]. The studies conducted in high-income countries have also shown that feeling healthy is one of the barriers to participate in a CCS program [75, 82, 83].

Women who had not received their test results from the HPV self-sampling pilot study and had not been offered follow-up and treatment for abnormal findings explained their dissatisfaction with their participation. Lack of clear communication between health care providers and clients, and logistic challenges were the reasons for not providing the test results to some of the participants. In turn, failing to provide test results on time and providing follow-up affected the acceptance of self-sampling in other studies as well [61, 74, 84].

The fear to apply the self-sampling instrument was the main barrier to the provision of quality HPV testing in the pilot study. Similarly, a study conducted in Cameroon noted that some of the participants reported feeling anxious to perform HPV self-sampling [72]. The challenges some of the participants faced could be related to the lack of proper training on how to use the self-sampling instrument. In addition, the low educational status, feeling shy to ask questions about the self-sampling procedure, sample collectors' inexperience of using Evalyn brush®, and failure of providing proper instructions might have contributed to this. The concern of not performing the self-sampling properly has been reported in other African countries [27, 31, 72]. Therefore, health education should use clear educational materials. Flipchart illustrations could help visualizing the procedure, and a sort of cultural-sensitive cartoon style could dispel fear.

The inability of the sample collectors to check the proper utilization of Evalyn brush® after use was found to be one of the reasons for the low quality of HPV testing. It seems that the study management of home-based HPV self-sampling wanted to help women to feel more comfortable by providing their closed samples, but it prevented the sample collectors from checking the proper utilization of the instrument. Hence, we suggest to develop a vagina model which could be used to demonstrate the procedure and to check the proper utilization of the device and at the same time keeps privacy of the women.

## Strength and limitation of the study

This study discovered the reasons behind the comparatively low acceptance rate and partly poor-quality samples of the HPV self-sampling intervention. In-depth interviews and focus group discussions were employed to collect data from four different study populations, this increases the triangulation. Female data collectors who were familiar with the culture of the community facilitated the discussion considering HPV self-sampling is a sensitive issue in the

community. In comparison with the qualitative study conducted prior to the HPV self-sampling, our study revealed additional barriers and some implementation failures.

One of the limitations of this study is that the same sample collectors of home-based HPV self-sampling conducted the in-depth interviews and facilitated the focus group discussions. This could have affected the participants' response in disclosing their challenges. Another limitation of this study is that men were not included in this study; thus, their opinions were only captured indirectly from the sample collectors. Furthermore, this study was conducted two years after the conduct of the community-wide home-based HPV self-sampling campaign; thus, some information could be affected by recall bias.

## Conclusion

This study identified a series of barriers to successful home-based HPV self-sampling. Husband disapproval, social influence, and lack of knowledge about cervical cancer and HPV testing were identified as the barriers to the acceptance of HPV self-sampling. The fear of applying Evalyn brush®, the inability of the sample collectors to check the proper utilization of Evalyn brush®, and the difficulty in understanding the procedure instructions limited the quality of the HPV testing sample. The findings of this study could help to improve the implementation of the program in similar settings of Ethiopian communities.

To improve the acceptability of self-sampling, we strongly recommend prioritizing the involvement of men in the CCS programs and the provision of proper training for women on how to collect the self-sample. Also, education and information sharing about home-based HPV self-sampling should be integrated into other reproductive health services to increase women's awareness about cervical cancer, screening, and to address the misconceptions in the community towards HPV self-sampling. Thus, as most of the rural women do not read and write, direct and verbal health education from the service provider should be given utilizing visual aid materials and a vaginal model.

In addition, women empowerment is needed to overcome barriers to acceptance of self-sampling such as cultural expectations of women to be modest, social and religious influence, and husband/partner disapproval to increase utilization of HPV testing. Moreover, the self-sampling kit (Evalyn brush®) should be available at nearest health facility (satellite health posts) and used for frequent health education e.g., during family planning education so that women could avoid the fear of using it.

Furthermore, community-based HPV testing should engage women, women's health development army leaders, community leaders, religious leaders, community health workers, and husbands in the screening program to effectively address the low cervical cancer screening in the study area. It is also very important to provide the test results on time and to offer follow-up and treatment for women with abnormal findings. Finally, future randomized studies could test certain measures, for instance, the participatory development and application of appropriate teaching materials.

## Supporting information

**S1 File.**
(PDF)

## Acknowledgments

We are grateful to the University of Gondar and Dabat research site for supporting us during data collection. We also thank all study participants for devoting their time to take part in this study.

## Author Contributions

**Conceptualization:** Bikila Soboka Megersa, Hermann Bussmann, Andreas Deckert.

**Data curation:** Bikila Soboka Megersa, Andreas Deckert.

**Formal analysis:** Bikila Soboka Megersa, Hermann Bussmann, Till Bärnighausen, Achenef Asmamaw Muche, Kassahun Alemu, Andreas Deckert.

**Funding acquisition:** Bikila Soboka Megersa.

**Investigation:** Bikila Soboka Megersa.

**Methodology:** Bikila Soboka Megersa, Hermann Bussmann, Till Bärnighausen, Achenef Asmamaw Muche, Kassahun Alemu, Andreas Deckert.

**Software:** Bikila Soboka Megersa.

**Supervision:** Hermann Bussmann, Andreas Deckert.

**Validation:** Till Bärnighausen, Achenef Asmamaw Muche, Kassahun Alemu.

**Writing – original draft:** Bikila Soboka Megersa, Hermann Bussmann, Andreas Deckert.

**Writing – review & editing:** Bikila Soboka Megersa, Hermann Bussmann, Till Bärnighausen, Achenef Asmamaw Muche, Kassahun Alemu, Andreas Deckert.

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
