## [Decision Letter · Decision Letter 0]

11 Jun 2020

PONE-D-20-06352

Community cervical cancer screening: Barriers to successful home-based HPV self-sampling in Dabat district, North Gondar, Ethiopia. A qualitative study.

PLOS ONE

Dear Dr. Megersa,

Thank you for submitting your manuscript to PLOS ONE. After careful consideration, we feel that it has merit but does not fully meet PLOS ONE’s publication criteria as it currently stands. Therefore, we invite you to submit a revised version of the manuscript that addresses the points raised during the review process.

We look forward to receiving your revised manuscript.

Kind regards,

Richard Hodge

Associate Editor

PLOS ONE

Journal Requirements:

2. Please provide additional details regarding participant consent. In the ethics statement in the Methods and online submission information, please ensure that you have specified (1) whether consent was informed.

3. In your Methods section, please provide additional information about the participant recruitment method and the demographic details of your participants. Please ensure you have provided sufficient details to replicate the analyses such as: a) the recruitment date range (month and year) and b) a description of any inclusion/exclusion criteria that were applied to participant recruitment.

4. Please provide a sample size and power calculation in the Methods, or discuss the reasons for not performing one before study initiation.

6. Please amend the manuscript submission data (via Edit Submission) to include authors Hermann Bussmann,

Till Bärnighausen, Achenef Asmamaw Muche, Kassahun Alemu and Andreas Deckert

Additional Editor Comments (if provided):

Reviewers' comments:

Reviewer's Responses to Questions

**Comments to the Author**

1. Is the manuscript technically sound, and do the data support the conclusions?

Reviewer #1: Yes

Reviewer #2: Partly

2. Has the statistical analysis been performed appropriately and rigorously? 

Reviewer #1: No

Reviewer #2: N/A

3. Have the authors made all data underlying the findings in their manuscript fully available?

Reviewer #1: No

Reviewer #2: No

4. Is the manuscript presented in an intelligible fashion and written in standard English?

Reviewer #1: Yes

Reviewer #2: No

5. Review Comments to the Author

Reviewer #1: Review PLOS ONE - PONE-D-20-06352

Community cervical cancer screening: Barriers to successful home-based HPV self-sampling in Dabat district, North Gondar, Ethiopia. A qualitative study.

This is a collaborative study between research groups in Ethiopia and Germany to assess the barriers to carrying out self-sampling of the vaginal canal at home to identify HPV as a cervical cancer screening strategy.

The main findings can be attributed ot sociocultural factors and indicate that the main barriers are the husband’s attitude, the social environment, lack of knowledge of cervical cancer and its etiology; and to absence of recommendation and acceptance of this practice by health workers as a whole and by localo community leaders.

The manuscript as such is a pilot study that uses a methodological strategy of in-depth interviews with 22 women and the development of 4 focus groups of 6-7 women each. It is widely described although the information is not very original and recommendations are few.

The information obtained is, again, not original. The results obtained are similar to those reported in the 1980s on barriers to the use of cervical cytology in poor countries, as well as on the acceptability of HPV testing and vaginal canal autotomy since the late 1990s.

Concern for the correct use of vaginal auto-collection instruments and the feeling of shame with male partners and health professionals is a fact of life in most cultures. The manuscript is generally very descriptive, with very limited recommendations for the context of a geographic area of very low cervical cancer screening coverage. Although the manuscript was written to describe barriers to utilization, it would have been advisable to list possible interventions to improve acceptability and how to address low healthcare coverage in population-based scree

Reviewer #2: Dear Authors, thank you for the opportunity to review your manuscript entitled “Community cervical cancer screening: Barriers to successful home-based HPV self-sampling in Dabat district, North Gondar, Ethiopia. A qualitative study”. This is a niece piece of work aimed at understanding women’s experience and perspectives about HPV self-sampling for cervical cancer screening following a pilot study in Ethiopia, Eastern Africa.

This paper could be improved by addressing the following concerns.

The abstract is not very well structured. For instance, the methods should start by describing the study population, then elaborate on the study type and design, before talking about the sampling approach. Also, the conclusion that “HPV self-sampling seems to be a promising strategy to increase low cervical cancer screening coverage in Ethiopia”, does not appear to be related to the study aims and findings.

In the data availability section of the manuscript, it says that some restrictions to data access may apply. Please could authors clarify why their data cannot be made publicly available?

The first paragraph of the introduction section of the manuscript is a bit confusing. On page 3 lines 42 to 43, it writes: “Cervical cancer ranks second among women aged 15+ years (3), whereas it is the most common cause of cancer in North Western Ethiopia (4)”. Do the authors refer to cervical cancer in terms of incidence, prevalence or mortality? What is the denominator for this comparison? Why would they compare women aged 15 years or older (I assume at the national level), with women (not sure what is the age range here) in North Western Ethiopia?

On page 3 lines 52 to 55, it writes: “HPV-based screening is considered to be cost-effective and to be the most suitable strategy to increase the coverage of CCS in a resource-poor setting due to better accessibility and enabling women to take a self-collected cervicovaginal sample at home (6, 9-15).” Do authors mean that HPV is more available than VIA in limited resource settings? Do they mean that self-collection is more valuable when performed at home? Are the reasons highlighted in this sentence the main reasons why HPV screening is the most cost-effective strategy for CCS in limited resource settings? Please, clarify.

On the other hand, many references used throughout the manuscript are not up-to-date. For instance, the current reference (6), could be replaced by more recent and more adapted reference,s such as the following: Fokom Domgue J & Valea F, 2018. Is it relevant to keep advocating VIA for cervical cancer screening in limited resource settings? Journal of Global Oncology, 2018; or Defo VF et al, 2020. Why consider self-sampling for cervical cancer screening in limited resource settings? AMA Journal of Ethics, 2020.

On page 4 lines 76 and 77, it writes: “Insufficient samples were provided in about one-fifth of the cases (publication in process, findings 77 similar to (19))”. This fact that HPV specimens and results were invalid or insufficient in about 20% of specimens, is not necessarily due to the poor quality of self-samples provided by women, but could be due to many factors related to the storage, transportation and processing of these specimens, the delay between self-sampling and HPV testing, inter alia. Please clarify. On the other hand, a 82.5% acceptance rate in a pilot study does not seem low to me, and the when comparing this rate to the existing literature, the authors should ensure that the study population in previous reports is comparable to theirs.

In the methods section, the main concern is related to the interval (about 2 years) between the implementation of pilot study (2017) and the conduct of the follow-up study presented in this manuscript (2019). Why did the authors wait for so long before surveying participants (including CHWs, sample collectors and women) after they were exposed to self-sampling? How did they ensure that the responses provided in 2019 about an experience they had in 2017 was accurate?

On the other hand, participants were classified according to whether their samples were sufficient or insufficient. As mentioned above, a sample classified as insufficient could be due to many factors other than the ability of women to properly perform self-sampling.

Further, the authors should check and correct grammatical and typo- errors that are found throughout the manuscript.

Once these points have been addressed, the results and the discussion can be updated accordingly.

Thank you.

6. PLOS authors have the option to publish the peer review history of their article (what does this mean?). If published, this will include your full peer review and any attached files.

Reviewer #1: No

Reviewer #2: Yes: Joel FOKOM DOMGUE

---

## [Author Response · Author response to Decision Letter 0]

15 Sep 2020

Rebuttal letter for Reviewers comment 

PONE-D-20-06352

Community cervical cancer screening: Barriers to successful home-based HPV self-sampling in Dabat district, North Gondar, Ethiopia. A qualitative study.

We thank the reviewers for the helpful comments. The changes made and our responses to the mentioned points are as follows:

Reviewer one 

Comment: It is widely described although the information is not very original, and recommendations are few. The results obtained are similar to those reported in the 1980s on barriers to the use of cervical cytology in poor countries, as well as on the acceptability of HPV testing and vaginal canal autotomy since the late 1990s.

Answer: In comparison to other studies, our study identified somewhat the real barriers to HPV self-sampling in the study area since the study conducted following the conduct of the preliminary community-wide home-based self-sampling. Yet, as commented by the reviewer, some of the findings of this study are similar to the findings of other studies conducted elsewhere to assess the acceptance of HPV self-sampling. However, some of the findings are rarely reported in other studies. For instance, husband and neighbour disapproval, religious influence as a barrier to the acceptance of HPV self-sampling, and some of the misconceptions about cervical cancer and screening identified in this study have been rarely reported in other HPV self-sampling studies. 

This study also identified some of the implementation limitations such as the inability of the sample collectors to check the proper utilization of self-sampling kit (Evalyn brush®) (page 20, lines 272-275), delay in providing test results, and follow-up and treatment for women with abnormal findings which are rarely reported in other studies (page 21, lines 289-294). Moreover, this study provided a clue on how it was difficult to understand the instruction of self-sampling for rural women which is rarely identified in other studies. Furthermore, the results of the study were used to develop a conceptual model to explain determinants for the success of HPV self-sampling which we did not come across in other studies (page 23, line 297-300). Thus, although our study shares some of the findings with other studies, still some of the findings are rarely reported. 

Comment: Concern for the correct use of vaginal auto-collection instruments and the feeling of shame with male partners and health professionals is a fact of life in most cultures. The manuscript is generally very descriptive, with very limited recommendations for the context of a geographic area of very low cervical cancer screening coverage.

Answer: As the reviewer has indicated, several studies have reported the concern of women’s ability to perform self-sampling correctly. However, our study did not identify women’s concern to perform the self-sampling correctly as a barrier, rather we identified that some of the women were afraid the instrument and thought it could hurt their genitalia during self-sampling (page 19, lines 259-265). Also, the woman’s feelings of shame with her husband/partner to attend cervical cancer screening service and to receive the service from health professionals have been widely reported from provider dependent, clinic settings where it involves pelvic examination. However, our study identified husband disapproval as the main barrier to acceptance of self-sampling. 

Regarding the recommendations to the study area, previously, we have listed some of the possible recommendations that could improve the acceptance and overall implementation of HPV self-sampling in the study area in the discussion and mainly under conclusion sections. In addition to the previous recommendations, we have added other recommendations in the conclusion part which could improve the acceptance of HPV self-sampling and the implementation of the screening program and consequently could rise the screening coverage in the study area. For example, we included women empowerment, increasing women’s familiarity with the self-sampling kit, and engaging all stalk holders in the screening program to tackle some of the identified socio-cultural barriers (pages 29, lines 428-440). 

Comment: Although the manuscript was written to describe barriers to utilization, it would have been advisable to list possible interventions to improve acceptability and how to address low healthcare coverage in population-based screening. 

Answer: As explained above, in addition to recommendations that have been provided, based on the comment, we have added other recommendations in the conclusion section that could help to improve the acceptance of HPV self-sampling and could help to overcome the low cervical cancer screening coverage in the study area (pages 29, lines 428-440). 

Reviewer two

Comment: The abstract is not very well structured. For instance, the methods should start by describing the study population, then elaborate on the study type and design, before talking about the sampling approach. Also, the conclusion that “HPV self-sampling seems to be a promising strategy to increase low cervical cancer screening coverage in Ethiopia”, does not appear to be related to the study aims and findings.

Answer: We have re-structured the abstract as commented by first describing the study population, study type and design, and thereafter continued with sampling technique. As the reviewer commented, the sentence “HPV self-sampling seems to be a promising strategy to increase low cervical cancer screening coverage in Ethiopia” in the conclusion part is not supported by the findings of the study and we have omitted the sentence. 

Comment: In the data availability section of the manuscript, it says that some restrictions to data access may apply. Please could authors clarify why their data cannot be made publicly available?

Answer: We have provided in-depth interviews and FGDs guides with quotations from IDIs and FGDs by themes in anonymized form as a supplementary file (Additional file 1).

Comment: The first paragraph of the introduction section of the manuscript is a bit confusing. On page 3 lines 42 to 43, it writes: “Cervical cancer ranks second among women aged 15+ years (3), whereas it is the most common cause of cancer in North Western Ethiopia (4)”. Do the authors refer to cervical cancer in terms of incidence, prevalence or mortality? What is the denominator for this comparison? Why would they compare women aged 15 years or older (I assume at the national level), with women (not sure what is the age range here) in North Western Ethiopia? 

Answer: It was the incidence of cervical cancer that ranks the second most common female cancer in Ethiopia. The reference [3] in the last version has reported the crude incidence of cervical cancer among the adult population of women, 15 years and above based on national cancer data registry, and the denominator was 100,000 population. However, the study that was conducted in Northwestern Ethiopia, reference [4] in the last version was a merely descriptive study reporting the pattern of all cancers by frequency and percentage from a cancer registry of a given hospital. Thus, we have removed it from the list citation. To avoid confusion, we omitted the sentence and replaced it with a clearer sentence “Cervical cancer ranks as the second leading female cancer for both incidence and mortality in Ethiopia (page 2, lines 42-43).” and we have cited the updated version of the Ethiopian human papillomavirus and related diseases report.

Comment: On page 3 lines 52 to 55, it writes: “HPV-based screening is considered to be cost-effective and to be the most suitable strategy to increase the coverage of CCS in a resource-poor setting due to better accessibility and enabling women to take a self-collected cervicovaginal sample at home (6, 9-15).” Do authors mean that HPV is more available than VIA in limited resource settings? Do they mean that self-collection is more valuable when performed at home? Are the reasons highlighted in this sentence the main reasons why HPV screening is the most cost-effective strategy for CCS in limited resource settings? Please, clarify.

Answer: “Do authors mean that HPV is more available than VIA in limited resource settings?” We do not mean that HPV is more available than VIA in resource limited setting. Although the WHO guideline recommends the use of either HPV testing or VIA for cervical cancer screening in LMICs, it indicates that VIA is more available for cervical screening than HPV testing in resource limited settings (https://www.ncbi.nlm.nih.gov/pubmed/25642554). In addition, the WHO expert panel recommends the use of VIA instead of HPV in resource constrained settings where HPV testing is not available (http://dx.doi.org/10.1016/j.ijgo.2015.07.038). Thus, we have revised the sentence to make it clear. “HPV-based screening is considered to be the most suitable strategy to increase the coverage of CCS in low-resource settings by enabling women to take a self-collected cervicovaginal sample for HPV testing at home.”

“Do they mean that self-collection is more valuable when performed at home?” We do not mean that self-collection is more valuable when it is done at home. Because as it has been indicated in some studies self-sampling could be performed at any place convenient to women including at home, workplace, and at health facilities. We did not come across any study comparing the quality of self-sampling performed at indicated places. However, we mean that home-based HPV self-sampling gives women the opportunity to perform self-sample at home which could increase the uptake of cervical cancer screening. As a result, it improves access to the screening for those who do not attend regular cervical cancer screening program and hard to reach rural women. 

 “Are the reasons highlighted in this sentence the main reasons why HPV screening is the most cost-effective strategy for CCS in limited-resource settings? Please, clarify.” We have stated the cost-effectiveness of HPV testing based on some studies which were included in the list of the references (6, 9-15). Some studies conducted in low- and middle-income countries have indicated that community-based self-collected HPV testing followed by treatment for HPV-positive women has the potential to be a cost-effective screening strategy (doi:10.1136/bmjopen-2017-020484).7). This is true particularly at high coverage levels and when self-collection is associated with greater coverage of screening-eligible women than provider-collection (doi:10.1093/heapol/czw182). In addition, a systematic review has shown that HPV self-sampling could be a cost-effective strategy in LMICs (https://doi.org/10.1016/j.ypmed.2019.105953). We have separated the sentence describing the cost-effective of HPV self-sampling and added new citations. “Also, studies conducted in low-income settings have shown that population based self-collected HPV testing followed by treatment for HPV-positive women has the potential to be cost-effective screening strategy.” 

Comment: On th e other hand, many references used throughout the manuscript are not up-to-date. For instance, the current reference (6), could be replaced by more recent and more adapted reference,s such as the following: Fokom Domgue J & Valea F, 2018. Is it relevant to keep advocating VIA for cervical cancer screening in limited resource settings? Journal of Global Oncology, 2018; or Defo VF et al, 2020. Why consider self-sampling for cervical cancer screening in limited resource settings? AMA Journal of Ethics, 2020.

Answer: Based on the comment, we have updated some of the references and cited the suggested references. 

Comment: On page 4 lines 76 and 77, it writes: “Insufficient samples were provided in about one-fifth of the cases (publication in process, findings 77 similar to (19))”. This fact that HPV specimens and results were invalid or insufficient in about 20% of specimens, is not necessarily due to the poor quality of self-samples provided by women, but could be due to many factors related to the storage, transportation and processing of these specimens, the delay between self-sampling and HPV testing, inter alia. Please clarify. On the other hand, a 82.5% acceptance rate in a pilot study does not seem low to me, and the when comparing this rate to the existing literature, the authors should ensure that the study population in previous reports is comparable to theirs.

Answer: As the reviewer commented, the insufficiency of the samples provided may not necessarily be confined to the inability of the women to collect sufficient self-sample. Additionally, other issues such as storage and transportation have partially could contribute to low quality samples. This, however, is unlikely since the study management followed strictly international guidelines and procedures to assure the quality of the collected samples https://doi.org/10.1016/j.pvr.2020.100198. First of all, each woman was told to use the Evalyn brush®, close it, and give back to the sample collectors immediately, then the sample was stored properly and transported to the storage site. The Evalyn brush™ sample was stored at room temperature (circa 25 °C) for a maximum of 8 weeks (https://doi.org/10.1016/j.pvr.2018.04.005) until processing and analysis. As it has been cited elsewhere, the samples were analyzed using the AID HPV-DNA array test (HPV Easy-Screening Kit from Autoimmun Diagnostika GmbH, Straßberg, Germany) (https://doi.org/10.1159/000502207, https://doi.org/10.1016/j.pvr.2020.100198 ). Quality control was done at the laboratory of the Clinic for Gynecology, Charité Universitätsmedizin Berlin, Germany using BSGP5+/6+ PCR followed by Luminex-MPG read-out (https://doi.org/10.1002/jmv.25583).

As such, after following the required procedures to assure the quality of the collected data, the adequacy of the sample for the HPV testing could merely be attributed to the quality of the self-sample collected from women. Similarly, several studies attributed the adequacy of the self-sample for HPV testing to the ability of the women to perform self-sampling (doi:10.4269/ajtmh.19-0045, DOI 10.1186/s13027-017-0123-z, doi: http://dx.doi.org/10.4314/ahs.v15i3.8). Also, in some of the literature samples which were not adequate returned to each woman and become satisfactory for HPV testing. 

Concerning the acceptance rate of the self-sampling at the study area, since the pilot study was the first in the country, we compared it with other studies conducted in other low-income settings as cited in the manuscript where the populations could be compared to ours. For instance, the acceptance rate of self-sampling in rural Senegal was 98.5% and 98% in rural Madagascar. The setting in which our study and those studies conducted is similar since the study participants were drawn from rural settings. 

Comment: In the methods section, the main concern is related to the interval (about 2 years) between the implementation of pilot study (2017) and the conduct of the follow-up study presented in this manuscript (2019). Why did the authors wait for so long before surveying participants (including CHWs, sample collectors and women) after they were exposed to self-sampling? How did they ensure that the responses provided in 2019 about an experience they had in 2017 was accurate?

Answer: The interval between the preliminary study and the current is attributed to the following reasons: First of all, the community based preliminary study was conducted in November 2017. The collected samples were sent for HPV testing to Germany and the sampling results were out in June 2018. Thereafter, the inception of the current proposal was proposed in November 2018. The proposal was developed over a period of 4 months and thereafter obtaining ethical clearance from the Heidelberg University and the University of Gondar took about 3 months. Thus, because of the above-mentioned reasons, the survey of CHWs, sample collectors, and women participated in self-sampling was conducted in 2019. 

Regarding the accuracy of the experiences of the participants on self-sampling, studies have demonstrated that recall of cervical cancer screening event is pretty accurate. A study has shown that 96% of cervical cancer screened patients according to their medical record accurately reported their screening https://www.ncbi.nlm.nih.gov/books/NBK539386/. In addition, the sample collectors had demonstrated the procedure of self-sampling to the participants which could help them to memorize the procedure and report their experiences. During the interview, women were also asked whether they remember the procedure of self-sampling, and the majority of those participated in the self-sampling were able to explain the procedure correctly. 

Comment: On the other hand, participants were classified according to whether their samples were sufficient or insufficient. As mentioned above, a sample classified as insufficient could be due to many factors other than the ability of women to properly perform self-sampling.

Answer: As described above the study management had taken internationally recognized procedures to assure the quality of the samples during collection, storage, transportation, and analysis https://doi.org/10.1016/j.pvr.2020.100198. Thus, the adequacy of the samples for the HPV testing could merely be attributed to the quality of the self-sample each woman provided. 

Comment: Further, the authors should check and correct grammatical and typo- errors that are found throughout the manuscript.

Answer: We have checked and corrected grammatical and typo- errors.

---

## [Editor Report · Decision Letter 1]

16 Nov 2020

Community cervical cancer screening: Barriers to successful home-based HPV self-sampling in Dabat district, North Gondar, Ethiopia. A qualitative study.

PONE-D-20-06352R1

Dear Dr. Megersa,

We’re pleased to inform you that your manuscript has been judged scientifically suitable for publication and will be formally accepted for publication once it meets all outstanding technical requirements.

Kind regards,

Eduardo Lazcano-Ponce

Guest Editor

PLOS ONE
---

## [Editor Report · Acceptance letter]

3 Dec 2020

PONE-D-20-06352R1 

Community cervical cancer screening: Barriers to successful home-based HPV self-sampling in Dabat district, North Gondar, Ethiopia. A qualitative study. 

Dear Dr. Megersa:

I'm pleased to inform you that your manuscript has been deemed suitable for publication in PLOS ONE. Congratulations! Your manuscript is now with our production department. 

Kind regards, 

on behalf of

Dr. Eduardo Lazcano-Ponce 

Guest Editor

PLOS ONE